# Perennial Semantic Data Terms of Use for Decentralized Web

## ABSTRACT

In today's digital landscape, the Web has become increasingly centralized, raising concerns about user privacy violations. Decentralized Web architectures, such as Solid, offer a promising solution by empowering users with better control over their data in their personal 'Pods'. However, a significant challenge remains: users must navigate numerous applications to decide which application can be trusted with access to their data Pods. This often involves reading lengthy and complex Terms of Use agreements, a process that users often find daunting or simply ignore. This compromises user autonomy and impedes detection of data misuse. We propose a novel *formal* description of Data Terms of Use (DToU), along with a DToU reasoner. Users and applications specify their own parts of the DToU policy with local knowledge, covering permissions, requirements, prohibitions and obligations. Automated reasoning verifies compliance, and also derives policies for output data. This constitutes a "perennial" DToU language, where the policy authoring only occurs once, and we can conduct ongoing automated checks across users, applications and activity cycles. Our solution is built on Turtle, Notation 3 and RDF Surfaces, for the language and the reasoning engine. It ensures seamless integration with other semantic tools for enhanced interoperability. We have successfully integrated this language into the Solid framework, and conducted performance benchmark. We believe this work demonstrates a practicality of a perennial DToU language and the potential of a paradigm shift to how users interact with data and applications in a decentralized Web, offering both improved privacy and usability.

## CCS CONCEPTS

• **Computing methodologies** → **Knowledge representation and reasoning**; • **Security and privacy** → *Usability in security and privacy*; • **Information systems** → *Semantic web description languages*.

## KEYWORDS

Decentralized Web, Data Terms of Use, Usage Control, Formal Modelling, Automated Reasoning, Notation 3

**ACM Reference Format:**
Anonymous Author(s). 2018. Perennial Semantic Data Terms of Use for Decentralized Web. In *Proceedings of Make sure to enter the correct conference title from your rights confirmation emai (Conference acronym 'XX).* ACM, New York, NY, USA, 12 pages. https://doi.org/XXXXXXX.XXXXXXX

## 1 INTRODUCTION

After years of development, the Web has become an indispensable part of people's life. However, the centralization of the Web has risen as a pressing challenge, leading to various issues like pervasive user behavioural manipulation, privacy breaches and an imbalance of power [34, 40]. Decentralization is viewed as a potential solution to address these issues [15, 21], and initiatives like Solid (Social Linked Data) [30] have gained attention for their aim to return *data and control* to the user, while respecting the openesss and fairness of Web standards and infrastructures.

In a decentralized Web, users store their data in their own storage (such as Solid Pods), and applications must request permission to use and store data there. This shift in data control has the potential to reduce 'vendor-lock-in', as it limits the privileged ownership of data currently prevalently observed in large platforms. Furthermore, it is also crucial to fostering competition among different applications. However, one aspect that has received less attention in the decentralized setting is *how users can sensibly decide which application should be granted permission to access their data*.

Assume in the decentralized setting, Alice wants to use a shopping app to buy shoes, which may require access to several types of data in her Pod, including shoe size, delivery address, and billing information. She is concerned about how such an app may handle her data ethically. In the meantime, Bob, the developer of a shopping app, HappyShop, wants to build trust with users and is willing to provide descriptions on how HappyShop handles users data through its 'Terms of Use'. However, users like Alice typically do not read these terms (aka. "the biggest lie on the Internet") because of information overload and their length [20, 24]. The problem is exacerbated in the decentralized setting with its numerous apps that may require users' decision regarding access to their data Pods. As a result, users autonomy may still be compromised in the decentralized Web, and can lead to many issues related to data misuse [8, 35, 39] or loss of trust [13].

Facing these challenges, we propose the concept of "perennial" Data Terms of Use (DToU), a formal language model that targets at addressing the following challenges in a decentralized Web context: 1) expressing data provider's DToU for their data; 2) imparting application's (developer's) DToU on how they handle data; 3) performing compliance checking over data usage requests; 4) supporting DToU policy reusing across applications and data providers; 5) facilitating apt DToU-compliant cross-application data sharing. With such a language model, automated reasoning can be performed thus only exposing distilled important information to users, reducing the amount of information and numbers of decisions exposed to the user, thus incentivizing responsible handling of Terms of Use. It is called "perennial" because a stakeholder only needs to specify the DToU once in the beginning, and it can be reused across activity cycles and across stakeholders, just like the plants being implanted once and kept growing in the future.

The *perennial* concept is in contrast to *sticky policy* [22, 26] which proposed principles for supporting distributed DToU compliance,

but, in the core definition, only explicitly discussed requirements 1 and 3. *Sticky policy* supports DToU-compliant cross-application data sharing, but suggests the policy staying the same regardless of data processing history, thus is not *apt*, leading to the potential of frequent user disruption. Existing research on policy languages, often coined as access control [27] and usage control [19, 32], also proposed formal models for expressing certain parts of DToU. There are different flavours of them, targeting at different scenarios, with diverse properties and capacities. Although with a different focus, they provide many design principles and concepts that are useful across contexts.

In this paper, drawing upon concepts from existing data policy languages, we propose a perennial policy language that addresses challenges from decentralization while maintaining expressiveness, by using heterogeneous yet interoperable data policy and application policy. The language and reasoning mechanism are built on semantic technologies, namely Turtle [5], Notation 3 [6] and RDF Surfaces [12]. This facilitates the creation of a common vocabulary, and enables integration with other semantic tools, particularly ontologies and ontological reasoning. Furthermore, our approach is integrated with Solid, a decentralized user-focused Web architecture, and we evaluate its performance across various workloads. To the best of our knowledge, we are the first work proposing a (semantic) perennial policy language that supports stakeholders expressing their DToU in the context of the decentralized Web. We believe this work provides a good starting of the paradigm shift for enhancing user autonomy and control.

## 2 RELATED RESEARCH

Several explorations have delved into the utilization of computer-interpretable formal encoding of policies to enable (semi-)automated decision-making of data usage authorization. These range from the classical access control to more advanced dynamic usage descriptions. In this section, we examine this body of research and discusses their relevance to a decentralized Web context, in addition to Table 1 which summarizes their main features.

The most well-known line of research involves various access control models, each based on different principles. For example, models such as Mandatory Access Control (MAC) [33], Access Control List (ACL), Role-Based Access Control (RBAC) [31] or Attribute-Based Access Control (ABAC), can be based on information ranging from narrower details like user identity to broader categories such as user groups, and contextual information. Several languages uses or implements them, such as E-P3P [18], WAC [4] and P2U [16]. There is also research that falls in the middle ground, such as Label-Based Access Control (LaBAC) [7], a simplified variant of ABAC while more expressive than MAC and RBAC, using simple tuples to express conditions on users, actions and data (objects).

Other policy languages may also use the concepts from access models, with additional features, such as eXtensible Access Control Markup Language (XACML) [1], a widely-known XML-based standard for ABAC with additional constructs like obligations for cloud services, and ODRL [2], which will be discussed later. In general, this type of research usually assumes a finite number of personnel (including applications) engaged in policy checking, and thus presupposes the presence of a 'supervisor' with abundant knowledge

to actively compile and maintain policies, particularly when there are personnel changes. While this assumption is reasonable for managing data usage at an institutional level, it presents challenges in contexts with multiple independent stakeholders that engage and disengage dynamically, as is the case in a decentralized Web.

Some other research also makes similar assumptions, but allows for customized mechanisms within their policies. For instance, P2U [16], a data-right language akin to a set of contracts, defines permitted data usage activities for applications, data, users and meta-information (e.g. "negotiable"). Thoth [9], on the other hand, uses its own logic-like policy language to express policies not only for read action, but also for write, update and declassification actions. It permits intricate evaluations of formulae that involve rich operators and the incorporation of external sources of information during policy evaluation. While these approaches vary in terms of expressiveness and usage contexts, they share a common limitation due to the assumption of a finite number of personnel. In our example, they can only define the policy *after* a user like Alice has determined whether a shopping app like HappyShop should be granted permission or not, rather than proactively.

In contrast to the approaches mentioned above, some research recognizes the dynamic nature of data and applications, and offers different solutions. LoNet [14], for example, employs Information Flow Control (IFC) for RBAC, along with special transformation rules for policy evolution. Additionally, it uses custom meta-code for checking additional policy information like time. However, the most expressive part, the meta-code, is arbitrary and difficult to statically verify. CamFlow [25], on the other hand, borrows concepts from Decentralized IFC (DIFC) [23], and employs homogeneous policy constructs (tags and labels) to encode the data policy and application capability, and checks their compatibility. It also allows the expression of policy evolution by adding or removing tags for output (policy) based on input (policy). Smart object [29] uses complex constructs to define permitted and prohibited use of data, and combines this with the application information (using relational calculus) to derive policies for output data, assuming and leveraging a tabular structure of data. Dr.Aid [28] focuses on the context of data-intensive scientific workflows, employing different structures to express data rules and process rules. It supports policy reasoning and derivation for workflow graphs involving multiple processes. A common feature among these approaches is the separation of data policy from application policy, using pre-defined reasoning mechanisms to facilitate compliance checking. They have demonstrated that this separation allows the reasoner to verify if an application complies with the data policy, aligning with our intended goal.

There are also policy languages that utilize semantic technologies or are tailored for specific use cases within the decentralized Web. One notable example is the Open Digital Rights Language (ODRL) [2], a well-known expressive policy language serializable to JSON-LD. It offers a comprehensive set of concepts, including permissions, prohibitions, obligations, remedies, conditions, purposes and agents. However, originated as a data right expression language, ODRL primarily focuses on expressing what is (not) permitted for the data, and lacks a corresponding mechanism for expressing application information. While efforts are being made to address this issue [10], there is still uncertainty regarding the eventual resolution. The AIR [17] policy language, based on Notation 3 (N3) [6],

**Table 1: Summarization of features under different categories of related policy languages.**

C1 - C5 refers to the Challenges 1 - 5 identified for perennial language. For column C1, A means authorization, O means obligation. For column C2, C means capacity, I means multi-input, M means use mode, P means purpose, T means data type/category. For column C3 & C4, T means true, T(e) means true if using same environmental information schema. For column C5, O means multi-output, T means transformation. For column Condition, A means application, C means capacity, E means (common) environmental information, **E** means Entire environmental information exposed in knowledge base, M means use mode, P means purpose, T means data type/category, U means user, X means external information.

| Language | C1 | C2 | C3 | C4 | C5 | Condition | Policy Author | Format |
|----------|-----|------|----|------|----|-----------|---------------|--------|
| E-P3P[18] | AO | | T | T(e) | | UEP | Supervisor | Custom |
| XACML[1]/ODRL[2] | AO+ | | T | T(e) | | EPU | Owner | XMl/JSON-LD |
| WAC[4]/ACP[3] | A | | T | | | AU/AUX | Owner | Turtle |
| P2U[16] | A | | T | | | AU | Owner | XML |
| LaBAC[7] | A | | T | | | AU | Supervisor | Custom |
| AIR[17] | A | | T | T(e) | | U**E** | Owner | N3 |
| Thoth[9] | A | | T | | | EUX | Owner | Logic-like |
| Eddy[8] | AO | MP | | T(e) | T | MPU | Supervisor | OWL-DL |
| LoNet[14] | A | | T | T | T | EU | Supervisor | Custom |
| CamFlow[25] | A | C | T | T | T | C | Owner | Custom |
| Smart object[29] | A | IPT | T | T | T | PT | Owner | Custom |
| Dr.Aid[28] | AO | IP | T | T | OT | EPU | Owner | Custom |
| **Ours** | AO | CIPT | T | T | OT | CEPU | Owner | Turtle |

specifies permitted and prohibited actions for data processing. It allows for the expression of custom rules directly on the contextual knowledge base of data processing. However, in the absence of general agreements across applications, this requires the policy author to have the knowledge of the information available in the knowledge base, resulting in a high level of coupling with the application. Eddy [8] examined real-life terms of use from online services and proposed a policy language based on OWL-DL to express key data requirements, including permissions, obligations and prohibitions. The language distinguishes between different types of data actions (collect, use, retain and transfer) and demonstrated compliance checking between two policy sets of two services. However, there is no clear demonstration of how data providers can utilize this language, as it necessitates highly detailed descriptions. Solid, a decentralized Web architecture based on Linked Data, has its own policy languages, mainly for access control purposes. This includes WAC [4], a policy language for expressing ACL, and ACP [3], an advanced policy language that supports a broad range of conditions, such as Verifiable Credentials. WAC and ACP leverage contextual information exposed by Solid but are less expressive compared to ODRL and AIR. In general, these languages cannot be directly applied to support data usage expressions in the decentralized setting we target at due to their individual limitations.

## 3 OUR LANGUAGE

### 3.1 Language design

Broadly speaking, our language model consists two parts: the data policy and the application policy. This design is crucial to enable automated data access negotiation between a data owner and a data consumer (e.g. applications). The *data* policy enables data providers to define policy-related metadata and expectations for the data consumers. The *application* policy allows application developers to encode the promises and expectations for accessing the data by the application.

The reasoner performs three types of tasks: a) conformance check: deciding whether the application can use the data; b) obligation check, assessing what obligations are activated by the application; and c) policy derivation: determining the policy for output data and saving a data owner to define data usage policy in every data access use case. Figure 1 gives an overview of the language and the relation between different concepts.

In the following, we provide more detailed descriptions about the language, with all examples expressed in Turtle [5]. For simplicity, we omit the prefixes, and only use : for named nodes.

*3.1.1 Data Policy.* Conceptually, the data policy consists of two layers: attributes and semantics. Attributes form the base layer, expressing *information* about the data and/or the policy. On top of that is the semantic layer, providing semantic concepts like tags, prohibitions and obligations, to support reasoning of information expressed in the attribute layer. In the Turtle syntax of the policy encoding, each tuple is expressed as a node with corresponding type and properties.

More precisely, each *attributes* is a simple tuple: (`name`, `class`, `value`). While they do not carry any semantics, they are designed with a flexible structure to describe information about data and policy. For example, attributes can be used to define Alice as an author, e.g. (`:author`, `:string`, `"Alice"`), or encode a textual description (e.g. (`:ack-text`, `:string`, `"This dataset is by Alice"`)), or specify the data fields (e.g. (`:col-2`, `:field`, `:column-2`)). This concept of *attributes* is borrowed from Dr.Aid [28], which has been demonstrated as a powerful structure for supporting policy derivation. The following example shows how Alice defines her email address information as an attribute – the attribute is a kind of 'string' and has the value of "alice@b.c."

```
:attr1 a :Attribute;
```

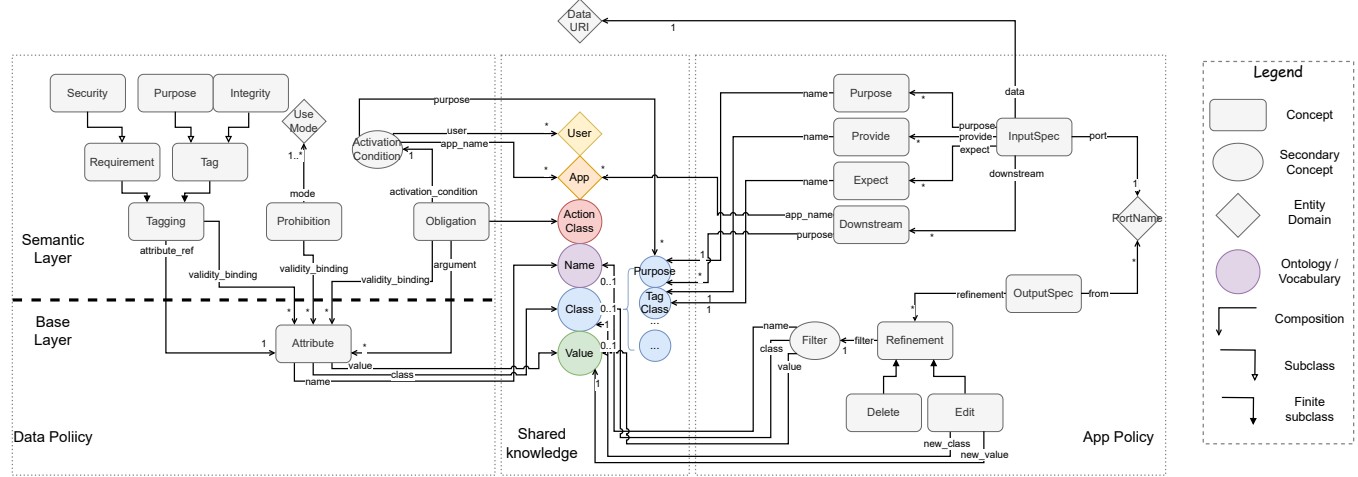

**Figure 1: Language design and relation between concepts**

```
:name :alice-email;
:class :string;
:value "alice@b.c".
```

*Tags*, from the semantics layer of our model, can be used to specify the requirements or available resources. Two typical types of tags are *security* and *integrity*: *security* specifies which security level(s) an application needs to possess to use the data; and *integrity* identifies the integrity level(s) this dataset has, to be requested by the application (policy). Inspired by the structure of CamFlow [25] (see Section 2), tags can be used to facilitate conformance check, see later Sec 3.2.2.

The name of a tag (i.e. which tag it is) is associated with the `class` of an *attribute* using `attribute_ref`; other information in the *attribute* are not used by a tag. Our policy language also allows expressing additional types of tags, such as *Purposes*.

The following shows how *tags* are used to define that Alice requires any application to respect `banking` security level, and permit `make-payment` and `verify-ownership` purposes (for payment info):

```
:attr-tag2 a :Attribute;
    :name :tag-2;
    :class :banking;
    :value :nil.
:attr-tag3 a :Attribute;
    :name :tag-3;
    :class :make-payment;
    :value :nil.
:attr-tag4 a :Attribute;
    :name :tag-4;
    :class :verify-ownership;
    :value :nil.

:tag2 a :SecurityTag;
    :attribute_ref :attr-tag2.
:tag3 a :PurposeTag;
    :attribute_ref :attr-tag3.
:tag4 a :PurposeTag;
    :attribute_ref :attr-tag4.
```

A *tag* may have *validity bindings*, meaning that the validity of the tag is dependent on the existence of referenced attribute(s). As an example, the tags in the payment info policy may have validity bindings to (an attributing denoting) the exact content of `:payment-details`:

```
:attr2 a :Attribute;
    :name :det;
    :class :data-content;
    :value :payment-details.

:tag2 a :SecurityTag;
    :attribute_ref :attr-tag2;
    :validity_binding :attr2.
:tag3 a :PurposeTag;
    :attribute_ref :attr-tag3;
    :validity_binding :attr2.
:tag4 a :PurposeTag;
    :attribute_ref :attr-tag4;
    :validity_binding :attr2.
```

*Prohibitions* specify additional restrictions on the data consumer, independent of the capacity (i.e. *tags*) of the application. The core of a prohibition is the *activation condition*, which is a matcher against user, application and purpose. If the condition matches the usage context, the usage is deemed prohibited. *Validity bindings* are also supported. Additionally, our model also offers an extension point, allowing users to specify the `:mode` when a match is considered. Currently, we only support the `:Use` mode, denoting the reading or processing of the data. We plan to explore other types of use modes to be integrated.

For example, Alice dislikes a payment processor, `<http://duckpay.com/>`, and does not want it to use her payment information, either directly or indirectly. This can be encoded as:

```
:pr1 a :Prohibition;
    :mode :Use;
    :activation_condition [
        :app_name <http://duckpay.com/>
    ];
    :validity_binding :attr2.
```

*Obligations* denote a potential obligation that will be triggered under certain conditions. Its core is an *obligation definition*, specifying what the triggered obligation will be, and an *activation condition*, specifying the condition to trigger this obligation. An *obligation*

*definition* contains an obligation class, and a list of arguments – references to attributes. Upon activation / instantiation of the obligation, the actual value of these attributes will be used, rather than the reference. The *activation condition* and *validity bindings* are the same as those defined for *prohibition*.

The following example shows that Alice expects any application to email her when it accesses her shoe size for research purposes:

```
:ob1 a :Obligation;
    :obligation_class :send-email;
    :args (:attr1);
    :activation_condition [ :purpose :research ].
```

*Policy set.* Finally, the policy terms explained above should be put together as a policy set. A policy set contains a `Policy` node which contains the relevant policy terms, and a `Data` node which pairs the policy and the data IRI that this policy applies to. For example, the policy set for Alice's payment information looks like:

```
:data-payment a :Data;
    :uri <http://a.b/payment-info>;
    :policy :policy-1.

:policy-1 a :Policy;
    :attribute :attr-tag2, :attr-tag3;
    :security :tag2;
    :purpose :tag3;
    :prohibition :pr1.
```

*3.1.2 Application Policy.* An *application policy* (abbreviated as *app policy*) contains basic information about the application, and the policy specification for the inputs and outputs. In addition, it also specifies relevant *downstream* data consumers (e.g. third-party APIs to send data to) that this application will use.

An application may have multiple inputs, and thus multiple *input specifications* (`:InputSpec`). Normally, each input specification describes basic information about that input (*name* and *data*) and its capacity: the *tags* it conforms to, including the *security* levels, the *integrity* it expects, and the *purpose* it will use the (input) data for. If the application sends the input data to an external location for processing (e.g. an API call), a *downstream* should be specified as a simplified app policy for that downstream stakeholder, specifying the (application) *name*, *user* and *purpose* of that *downstream*[1].

For example, Bob states this input specification for reading the payment information for HappyShop:

```
:input1 a :InputSpec;
    :data <http://a.b/payment-info>;
    :port "payment-info-in";
    :security :banking;
    :purpose :making-payment;
    :downstream [
        :app_name <http://goodpay.com/>;
        :purpose :making-payment
    ].
```

It says an input port named `"payment-info-in"` reads data from `<http://a.b/payment-info>`, and promises to comply with security level `:banking`, and will use the data only for purpose of `:making-payment`; it will send the data to a downstream, named `<http://goodpay.com/>`, for purpose of `:making-payment`. It is compatible with Alice's data policy for payment info.

---

[1]The app developer should verify that the downstream capacity is in line with that of the input specification, so its tags do not need to be explicitly expressed.

Similarly, an application may wish to store data into user's Pods, so it may need to specify multiple *output specifications*. Apart from the *name*, each *output specification* describes the related input that this output data is derived *from*, and the *refinements* that the data policies are subject to.

The *from* statement, (during reasoning) associates the output with a set of data policies, each pertinent to the input used to derive the output. In a higher level, this reflects the general information flow in the application. The removal or change of information is captured by a *refinement*, which expresses how (the *attributes* of) the data policies should be modified to reflect the processing that has been applied to the data. Two types of *refinements* are supported: *delete* and *edit*. A *delete* has a *filter*, meaning that all attributes matching the *filter* should be deleted, and the *filter* is used to specify the matching *attribute* information: *name*, *class* and *value*. Similarly, an *edit* means that any *attribute* matching the filter will be assigned a new class and value.

The following example output specification shows how HappyShop may record purchase histories in their Pod, which contains the copy of the delivery address, and a declassified version of payment details:

```
:out1 a :OutputSpec;
    :from "address-in", "payment-info-in";
    :refinement :refine-no-payment-details.

:refine-no-payment-details a :Delete;
    :filter [
        :class :data-content;
        :value :payment-details
    ].
```

This policy snippet means this output is related to the data from the inputs named `"address-in"` and `"payment-info-in"`, and has one refinement, which will delete all attributes of class `:data-content` and value `:payment-details`, reflecting a fact that the output data will not contain payment details even though it uses payment data.

The *refinements* directly operate on *attributes*, but references to attributes (*bindings*) in the semantic layer ensure that all operations will be inferred to related semantic concepts too. For instance, if an *attribute* is deleted, any *tags*, *prohibitions* and *attributes* that has a *binding* to this *attribute* will also be deleted.

*3.1.3 Shared vocabulary.* It is worth noting that shared vocabularies are assumed in all related research, with different levels of difficulties to achieve. In our work, this is achieved by making concepts explicit and using IRIs. This allows easy and decentralized provision of them, such as using OWL ontologies [38] as vocabularies. There are five sorts of vocabularies to be shared, as seen in the middle of Figure 1. They are centred around the data provider's wills, and thus is most natural to be provided by them, or some intermediaries. We mainly identify this mechanism, and leave this to the practitioners to consolidate the vocabularies. Relatedly, OWL reasoning may be integrated and performed before policy reasoning to maximize interoperability.

## 3.2 Reasoning

Our language accommodates three types of reasoning tasks: conformance check, obligation check, and policy derivation. This section explains the reasoning mechanism in more details.

In general, the reasoning rules can be expressed using first-order logic, encoded using RDF Surfaces [12] and Notation 3 (N3) [6] in our implementation. N3 is a language supporting the expression of both semantic data and reasoning rules with a rich syntax; RDF Surfaces, building on N3, provides a (direct) translation of first-order logic (FOL), including representing negations. Some of these rules contain explicit negations, which are implemented using N3 built-ins like `log:collectAllIn`, enabling scoped negation-as-failure (SNAF) [6]. For the sake of brevity, we only introduce the foundational principles here, and present the axioms in Appendix A.

*3.2.1 Context preparation.* Before embarking on the three reasoning tasks, it is essential to inject the contextual information and link the application policy with data policy.

The contextual information is represented as a *UsageContext*, specifying the relevant application policy *app_pol*, the *user* and the *time* of data usage. These parameters are only known during actual application requests and should be provided to the reasoner dynamically each time. Each reasoning task involves a distinct*UsageContext*.

Furthermore, all relevant policy content is added to the same knowledge base. This leverages the fact that Turtle is a sub-language of N3, making all policy specifications in Turtle valid N3 statements. The linkage between data policy and app policy is established by identifying the data URIs as specified in their respective fields. This is achieved through our reasoning rules, removing the need for additional injection of data policy into the corresponding inputs.

*3.2.2 Conformance check.* The fundamental purpose of our policy language is to determine whether a data usage should be permitted, which is essentially conformance checking. In our language, there are three types of conflicts to be checked: unsatisfied requirements, unmatched expectations, and prohibited uses.

An *unsatisfied requirement* occurs when a *requirement tag* (e.g. security) in the data policy is absent in the app policy. Therefore, the reasoning process involves determining all corresponding inputs and data, and verifying their requirement tags.

Conversely, an *unmatched expectation* arises when a *tag expectation* (e.g. integrity) in the app policy is missing in the data policy. This task also covers the verification of whether all purposes are permitted. Both the reasoning about *unsatisfied requirement* and *unmatched expectation* require a 'close-world assumption,' as it's essential to establish whether a tag does not exist. We utilize scoped negation-as-failure (SNAF) provided by N3 built-ins, considering the policy documents as the sole reliable source of information.

A *prohibited use* is identified when a prohibition is triggered. Prohibitions have their own semantics, including activation conditions, and should be checked in accordance with these conditions.

*3.2.3 Obligation check.* Our policy language can also reason about the obligations triggered during data use. This process is similar to checking prohibitions and involves verifying the *activation conditions*. But obligations contain arguments that are references to attributes and these attributes need to be returned from the query for further assessment.

*3.2.4 Policy derivation.* When the application produces output data (i.e. storing data to users' Pods), policy derivation becomes crucial to produce derived data policy for the output, based on the output specification. This aspect is central to (addressing challenge 5 of)

the *perennial* nature of our language. Policy derivation involves merging the data policies from all corresponding `from` inputs and performing *refinements*.

The derivation of *output attributes* is a primary focus because these attributes are vital for handling output policies for the semantics layer, particularly for bindings. Because of *refinements*, each input attribute will either have a copy, be edited, or cease to exist in the output. This process creates new nodes for the output attributes in the RDF graph, corresponding to the existential quantifier in the conclusions. The linkage between the input and output attributes is also recorded, which will be used for policy derivation of the semantic layer. SNAF plays a crucial role here as it determines what 'happens' to the rest of the input attributes that do not have a matching *refinement* – there should be an output attribute with identical name, class and value.

The *tags* in an output are based on the collection of all tags of related inputs, while removing those tags with deleted attribute bindings. Because tags have types, they can be treated uniformly when reasoning about their existence, and the reasoner can make use of the types afterwards.

The *output obligations* and *output prohibitions* are derived similarly. We first check if any binding is deleted, like for *output tags*. If not, a new node for obligation (or prohibition) is created, replicating all fields of the original obligation (or prohibition) in input. For obligations, that covers the obligation definition (obligated action class and arguments), validity binding and activation condition; for prohibitions, that covers the use mode, validity binding and activation condition.

Intuitively, the attribute references for the output policies will be the corresponding output attributes, instead of the input attributes.

## 4 SOLID INTEGRATION

To test and demonstrate the language, we integrated the language into Solid, a decentralized Web architecture based on Linked Data that emphasizes on user autonomy. This allowed us to express *data policies* and *application policies* and perform reasoning, in a realistic context, and also serving as the foundation for our benchmark.

To achieve this, we extended the Community Solid Server [11] v6.0, a modular and extensible Solid server implementation written in TypeScript. For policy reasoning, we use an off-the-shelf reasoner, EYE [36], which supports RDF Surfaces and N3 reasoning. In particular, we used the `eyereasoner` package available on npm[2], which is a WebAssembly distribution of EYE with a JavaScript interface.

In the sequence diagram depicted in Figure 2, we illustrate the key components and actions related to policy reasoning for applications, assuming that the data already have DToU policies associated.

Before reasoning, the application needs to register its application policy. The DToU handler, located behind the API endpoint on the server, processes the request and creates a temporary policy record.

Subsequently, the application requests a **conformance check** before utilizing the data. The DToU handler retrieves the corresponding policy, establishes the `UsageContext`, identifies the input data from the application policy, retrieves the relevant data policy, and calls the policy engine to perform the conformance check. The results are then provided to the application for further action, such

---

[2]https://www.npmjs.com/package/eyereasoner

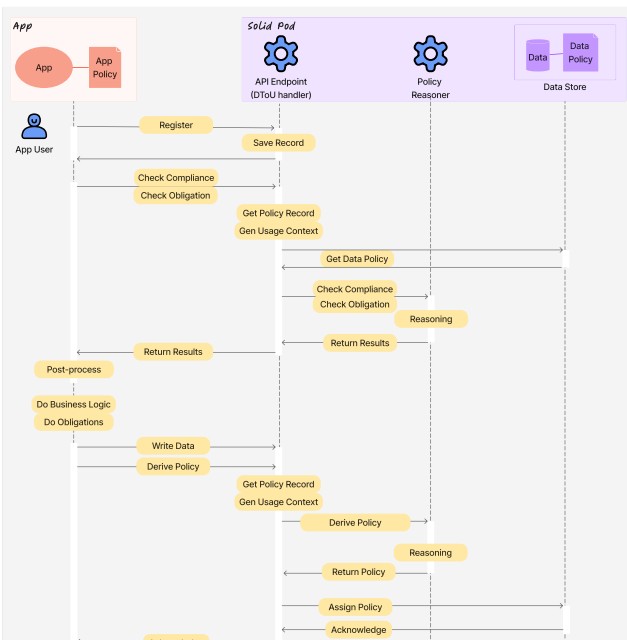

**Figure 2: Sequence diagram for the Solid integration of the DToU language**

as user display. Optionally, the DToU handler may restrict data access request by the application, in the presence of conflicts, or when data lacks a policy.

Similarly, the application can request an **obligation check** and then processes the information accordingly, including displaying it to the user at the appropriate time or automatically fulfilling obligations when applicable. Our language distinguishes between `UserObligation` and `ProcessObligation` classes to facilitate this distinction, leaving the specific obligations up to policy authors.

Finally, when the application intends to write output data to the Pod, it should send a **policy derivation** request along with the data. The DToU handler will then perform policy derivation and store the policy with the data. This allows subsequent applications to automatically carry out DToU reasoning when they request this data as input. As mentioned, the DToU handler may deny usage of data without policies, thus effectively enforcing policy derivation – if it does not perform it, the stored data will not be usable.

DToU reasoning is performed by the policy engine and DToU handler in the modified Solid service, reducing the burden on the application developers and Pod owners. From the Pod owner's perspective, supporting DToU requires expressing relevant data policies for input data. For the applications, DToU support involves preparing app policies and sending policy-related requests to the (modified) Solid service while handling responses. These application policies can be statically attached to the project, or dynamically generated from a template, offering flexibility for different users. The optimal method for authoring app and data policies remains an open question, outside the scope of this paper.

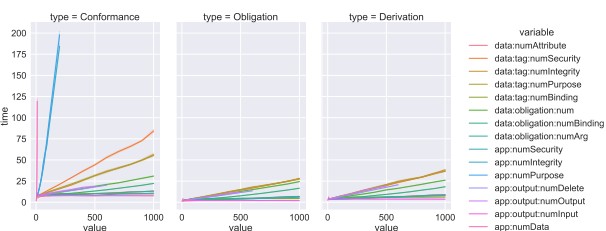

**Figure 3: General benchmark results of different workloads**

## 5 PERFORMANCE BENCHMARK

We conducted benchmark tests to evaluate the performance of our integration and to gain insights into its scalability across different reasoning tasks and workloads. This section presents the results and our discussion.

### 5.1 Benchmark settings

The benchmark encompasses a wide range of workloads of incorporating key variables: the number of different terms in the data policy and app policy. We varied these numbers, ranging from 10 to 1000, while keeping other variables at a fixed value of 10. There are two exceptions that are not fixed to 10: the number of attributes, which is set to 100, and the number of inputs, which is limited to 4. We chose these values because attributes require references to them, and handling only 10 would not suffice for distribution among tags, prohibitions and obligations; the number of inputs significantly impacts performance, so we opted for a smaller yet reasonable number.

It is worth noting that the range we benchmarked should be demonstrable to what usually exists in current real-world policies. For example, [8] reviewed Facebook, Zynga and AOL policies and identified a total of 131, 190 and 75 statements. Although not equitable, each statement roughly corresponds to one term in the semantic layer and several attributes, or one term in the app policy.

For the benchmark, we utilized the WebAssembly distribution of the eyereasoner v6.9.5, which is available on NPM. We started the server using the provided file-storage configuration, without special parameters. The benchmark tests were conducted on a consumer-level laptop, with an Intel Core i5-1135G7 (2.4 GHz) and 32GB of RAM, running on Linux kernel 6.1.55 (x86_64). Each workload was repeated 10 times, and the time taken from sending requests to receiving results was recorded.

### 5.2 Results and discussions

*5.2.1 General results.* Figure 3 shows the primary results of our benchmark. Most variables exhibit a linear or sublinear scaling trend. However, some variables do not reach 1000 because they encountered time-out or connection resets with further increase.

The three specific variables, namely `app:numData`, `app:numPurpose` and `app:numIntegrity`, displayed exceptional growing trends in conformance checking, resulting in time-outs, which we will discuss

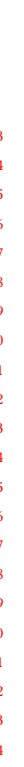

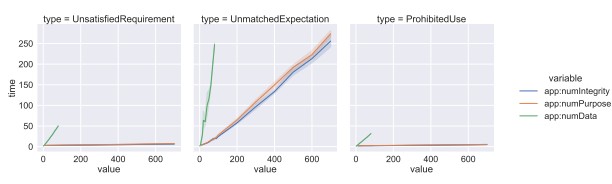

**Figure 4: Selective benchmark results for most variables, subtracting the base reasoning time for general axioms**

**Figure 5: Experiment results for different conformance checking tasks for exceptional variables, `app:numData`, `app:numPurpose` and `app:numIntegrity`**

separately later. After removing these exceptional cases, and subtracting the base time for policy loading and general reasoning (e.g. axioms for `rdfs:subClassOf`), we obtain Figure 4. It demonstrates a clearer linear growth trend in the remaining variables.

Among them, most variables reached 1000, except for `app:output:numDelete` and `app:output:numOutput`, which stopped at 600 and 500. This limitation was due to connection resets, caused by the server encountering JavaScript out-of-heap errors during the 10 repetitions. This indicates a potential area for optimization of the memory consumption. However, it's worth noting that having 500 output ports or 600 delete refinements per output port (thus 6000 in total) for an application is exceptionally large and likely not an issue in realistic scenarios.

*5.2.2 Unmatched expectations.* Now, regarding the exceptional growths, as shown in Figure 3, two variables, `app:numPurpose` and `app:numIntegrity`, correspond to the checking of unmatched expectations. We further performed additional experiments to understand the time spent on different conformance checking (sub-)tasks, as shown in Figure 5. It verified the intuition that the time was mainly consumed by checking the unmatched expectations.

However, it's interesting to note that the time for its symmetric task, unsatisfied requirements, did not exhibit the same exceptional growth, as seen in Figure 3 for variable `data:tag:numSecurity` and `app:numSecurity`. This suggests that the issue is likely not due to a mis-implementation of our axioms, but rather a performance bottleneck for RDF Surfaces or the EYE reasoner related to specific rule combinations and orders. Ideally, future work should explore and address this issue, with the potential to further optimize by reducing the coefficient for tag matching.

*5.2.3 Number of data / inputs.* Changing the number of data inputs led to the most significant growth in reasoning time, displaying a

growth pattern that appears higher than linear, as shown in the sub-figure for unmatched expectations in Figure 5. This could be explained because all reasoning tasks depend on the pairing of data policies and inputs, making the complexity increase faster with the number of data inputs. However, it's interesting to note that obligation checking and policy derivation did not pose the same exceptional growth trend as conformance checking in Figure 3. A plausible explanation is that conformance check involves a heavier workload, making this effect more pronounced. This is supported by the fact that the time spent on a more complex subtask, such as unmatched expectation, also grew faster than that of unsatisfied requirements and prohibited use, as reflected in Figure 5. This suggests that our usage of RDF Surfaces may be suboptimal and should be addressed and optimized in future work.

### 5.3 Conclusion

From our benchmark, it is clear that the reasoning cost for all tasks shows a linear growth concerning all but one factor. It is important to note that DToU policy reasoning is not performed frequently (typically three times in an application's lifecycle), so real-time performance is not a strict requirement. However, our result highlight the significant potential for deploying our DToU language on a large scale with a substantial volume of policies. Additionally, we delved into the specifics of the suboptimal results and proposed potential reasons for their behaviour. In a production system, it is essential to focus on optimizing the reasoning tasks and the underlying reasoner to ensure efficient and reliable performance.

## 6 SUMMARY AND FUTURE WORK

In this work, we have undertaken a comprehensive exploration of the challenges and advantages associated with introducing DToU into a decentralized Web context, exemplified by Solid, with the overarching aim of enhancing user autonomy. Our efforts have included identifying the pertinent challenges and benefits, delineating the specific requirements for a policy language within this context, and introduced our own DToU policy language, which is based on semantic technologies. We have also detailed the design of data policies, application policies and the underlying reasoning mechanism. Furthermore, we showed how our solution integrates with Solid, along with benchmark tests that assessed the scalability of our implementation, highlighting its potential for wider adoption with a substantial volume of policies. We discussed areas where further optimization is required.

The next step of our work involves evaluating the language expressiveness and its understandability for users. We are also interested in exploring simpler methods for policy authorization, which could potentially involve leveraging NLP technologies. In general, our work underscores a paradigm shift in how application may be selected, permission granted, and interoperability achieved. This shift is driven by the automated reasoning of *perennial* DToU policies. It points out a wide spectrum of challenges and opportunities, including but not limited to enhancing the language expressiveness, improving usability, effectively maintaining policies, and optimizing performance. Effectively addressing these social-technical challenges will require interdisciplinary collaboration, and we are committed to contributing to this ongoing effort.

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

# A  AXIOMS FOR REASONING

The axioms for reasoning are encoded as RDF Surfaces in our reasoner. For simplicity, we present the equivalent first-order logic

axioms for performing reasoning for our language here. Because our policy language is based on Turtle thus RDF [37], knowledge is represented as triples in the ABox. We employ some conventions here for the expression in first-order logic:

(1) $A(x, SomeType)$ denotes that the type of entity $x$ is $SomeType$, i.e. the triple $(x, rdf : type, SomeType)$ exists in the ABox;

(2) $hasPredicate(x, y)$ denotes that there is a relation $predicate$ between entity $x$ and $y$, i.e. the triple $(x, predicate, y)$ exists in the ABox;

(3) Other predicates starting with a capital letter refer to a shorthand, which will be explained individually;

(4) Variables are universally quantified over the scope of the entire formulae, unless explicitly quantified or stated otherwise;

(5) For clarify, we write the conclusion in the beginning, similar to a Horn clause;

(6) Existential quantified entities in the conclusions denote a new node in the ABox, which is handled nicely by RDF Surfaces;

(7) Negations denote scoped negation-as-failure, implemented using N3's built-ins;

(8) We use double-line arrows to visually distinguish between the main premise and conclusions; however, they still represent material implications, same as the single-line arrows, and are encoded in RDF Surfaces as such;

(9) Namespaces are dropped from the formulae for clarity.

## A.1 Helper axioms

To simplify the axioms for actual reasoning tasks, we extract two common parts as helper axioms: $RelatedDataAppInput$ and $InputPolicyForOutput$.

The $RelatedDataAppInput(usage, data, app, input)$ identifies and groups data policy ($data$) and corresponding input specification ($input$) together, as well as the usage context $usage$ and application policy $app$. Typically, in a reasoning, there is only one usage context ($usage$) and one application ($app$). Conformance checking often uses this predicate. It is defined as:

$$RelatedDataAppInput(usage, data, app, input) \Leftarrow$$
$$A(usage, UsageContext)$$
$$\wedge hasApp(usage, app_s) \wedge hasPolicy(app_s, app)$$
$$\wedge hasInputSpec(app, input) \wedge A(data, Data) \wedge hasUri(data, uri)$$
$$\wedge hasData(input, uri)$$

The $InputPolicyForOutput(input, policy, output)$ identifies which input (specification) and its corresponding data policy is related to an output (specification). One input has only one data policy, while one output may have multiple inputs, leading to several $InputPolicyForOutput$s. It is defined as:

$$InputPolicyForOutput(input, policy, output) \Leftarrow$$
$$RelatedDataAppInput(usage, data, app, input) \wedge$$
$$hasOutputSpec(app, output) \wedge hasFrom(output, inputPort) \wedge$$
$$hasPort(input, inputPort) \wedge hasPolicy(data, policy)$$

## A.2 Conformance check

We have explained them in the main text of the document. Therefore, this part only presents the formulae, and their explanations where necessary.

### A.2.1 Unsatisfied requirement.

$$UnsatisfiedRequirement(t, n, input) \Leftarrow$$
$$RelatedDataAppInput(usage, data, app, input) \wedge$$
$$hasPolicy(data, pol) \wedge hasRequirement(pol, req) \wedge$$
$$hasType(req, t) \wedge hasName(req, n) \wedge$$
$$\neg \exists prov.(hasProvide(input, prov) \wedge$$
$$hasType(prov, t) \wedge hasName(prov, n))$$

The definition of $UnsatisfiedRequirement(t, n, input)$ is:

$$UnsatisfiedRequirement(t, n, input) \equiv$$
$$\exists x.A(x, UnsatisfiedRequirement) \wedge hasType(x, t) \wedge$$
$$hasName(x, n) \wedge hasInput(x, input)$$

### A.2.2 Unmatched expectation.

$$UnmatchedExpectation(t, n, input) \Leftarrow$$
$$RelatedDataAppInput(usage, data, app, input)$$
$$hasPolicy(data, pol) \wedge hasExpect(input, exp) \wedge$$
$$hasType(exp, t) \wedge hasName(exp, n) \wedge$$
$$\neg \exists tag.(hasTag(pol, tag) \wedge hasType(tag, t) \wedge hasName(tag, n))$$

The definition of $UnmatchedExpectation(t, n, input)$ is:

$$UnmatchedExpectation(t, n, input) \equiv$$
$$\exists x.A(x, UnmatchedExpectation) \wedge hasType(x, t) \wedge$$
$$hasName(x, n) \wedge hasInput(x, input)$$

### A.2.3 Prohibited use.

$$ProhibitedUse(m, n, p, input) \Leftarrow$$
$$RelatedDataAppInput(usage, data, app, input) \wedge$$
$$hasPolicy(data, pol) \wedge hasProhibition(pol, pro) \wedge$$
$$hasMode(pro, Use) \wedge hasActivationCondition(pro, ac) \wedge$$
$$hasApp(ac, n) \wedge hasPurpose(ac, p) \wedge ($$
$$(hasName(app, n) \wedge hasPurpose(input, p)) \vee$$
$$(hasDownstream(input, ds) \wedge hasAppName(ds, n) \wedge$$
$$hasPurpose(ds, p))$$
$$)$$

The definition of $ProhibitedUse(m, n, p, input)$ is:

$$ProhibitedUse(m, n, p, input) \equiv$$
$$\exists x.A(x, ProhibitedUse) \wedge hasMode(x, m) \wedge hasName(x, n) \wedge$$
$$hasPurpose(x, p) \wedge hasInput(x, input)$$

## A.3 Obligation check

$ActivatedObligation(ob, args, input) \Leftarrow$
$\quad RelatedDataAppInput(usage, data, app, input) \land$
$\quad hasPolicy(data, pol) \land hasObligation(pol, obl) \land$
$\quad hasObligationClass(obl, ob) \land hasArgs(obl, args) \land$
$\quad hasActivationCondition(obl, ac) \land hasUser(ac, U) \land$
$\quad hasApp(ac, N) \land hasPurpose(ac, P) \land$
$\quad hasUser(usage, U) \land hasName(app, N) \land hasPurpose(input, P)$

The definition of $ActivatedObligation(ob, args, input)$ is:

$ActivatedObligation(ob, args, input) \equiv$
$\quad \exists x.A(x, ActivatedObligation) \land hasObligationClass(x, ob) \land$
$\quad hasArgs(x, args) \land hasInput(x, input)$

Slightly different from the conflicts, when querying activated obligations, the corresponding attributes should be returned as well.

## A.4 Policy derivation

### A.4.1 Output attribute.

$OutputAttribute(n, t, v, p, attr) \Leftarrow$
$\quad InputPolicyForOutput(input, policy, output) \land$
$\quad hasPort(output, p) \land hasPort(input, port) \land ($
$\quad (hasAttribute(policy, attr) \land hasName(attr, n) \land$
$\quad hasClass(attr, t) \land hasValue(attr, v) \land \neg \exists refi, filter.$
$\quad hasRefinement(output, refi) \land hasFilter(refi, filter) \land$
$\quad hasInput(filter, port) \land hasName(filter, n) \land$
$\quad hasClass(filter, t) \land hasValue(filter, v)) \lor$
$\quad (hasAttirbute(policy, attr) \land hasName(attr, n) \land$
$\quad hasClass(attr, t') \land hasValue(attr, v') \land$
$\quad hasRefinement(attr, refi) \land A(refi, Edit) \land$
$\quad hasFilter(refi, filter) \land hasInput(filter, port) \land$
$\quad hasName(filter, n) \land hasClass(filter, t') \land$
$\quad hasValue(filter, v') \land hasNewClass(refi, t) \land$
$\quad hasNewValue(refi, v))$
$\quad )$

Same as above, $OutputAttribute(n, t, v, p, attr)$ is a shorthand. However, different from them, it involves multiple nodes:

$OutputAttribute(n, t, v, p, attr) \equiv$
$\quad \exists attr'.A(attr', Attribute) \land hasName(attr', n) \land \land$
$\quad hasClass(attr', t) \land hasValue(attr', v) \land$
$\quad \exists fl.A(fl, ForwardLink) \land hasOrigin(fl, attr) \land$
$\quad hasPort(fl, p) \land hasRef(fl, attr')$

This means that we create a node $attr'$ which is of type $Attribute$, and assign its corresponding fields; we also create a linking relation (the node $fl$) between the original attribute $attr$ and the created attribute node $attr'$ at output $P$.

In addition to that, because we will often refer to the output attribute, we define this shorthand:

$IOPair(in, out, p) \equiv$
$\quad A(fl, ForwardLink) \land hasOrigin(fl, in) \land$
$\quad hasPort(fl, p) \land hasRef(fl, out)$

### A.4.2 Output tag.

$OutputTagging(t, ar, p, tag) \Leftarrow$
$\quad InputPolicyForOutput(input, policy, output) \land$
$\quad hasPort(output, p) \land hasTagging(policy, tag) \land hasType(tag, t) \land$
$\quad hasAttributeRef(tag, ar_0) \land IOPair(ar_0, ar, p)$
$\quad (\forall vb.hasValidityBinding(tag, vb) \rightarrow$
$\quad \quad \exists attr.IOPair(vb, attr, p))$

where the full form of $OutputTagging$ is:

$OutputTagging(t, ar, p, tag) \equiv$
$\quad \exists ot.A(ot, Tagging) \land hasType(ot, t) \land$
$\quad hasAttributeRef(ot, ar) \land hasPort(ot, p) \land$
$\quad (\forall vb, attr.hasValidityBinding(tag, vb) \land IOPair(vb, attr, p)$
$\quad \rightarrow hasValidityBinding(ot, attr))$

### A.4.3 Output prohibition.

$OutputProhibition(m, ac, p, pr) \Leftarrow$
$\quad InputPolicyForOutput(input, policy, output) \land$
$\quad hasPort(output, p) \land hasProhibition(policy, pr) \land$
$\quad hasUseMode(pr, m) \land hasActivationCondition(pr, ac) \land$
$\quad \forall vb.(hasValidityBinding(pr, vb) \rightarrow \exists attr.IOPair(vb, attr, p))$

where the full form of $OutputProhibition$ is:

$OutputProhibition(m, ac, p, pr) \equiv$
$\quad \exists op.A(op, Prohibition) \land hasUseMode(op, m) \land$
$\quad hasActivationCondition(op, ac) \land hasPort(op, p) \land$
$\quad \forall vb, attr.(hasValidityBinding(pr, vb) \land IOPair(vb, attr, p)$
$\quad \rightarrow hasValidityBinding(op, attr))$

### A.4.4 Output obligation.

$OutputObligation(oc, arg, ac, p, ob) \Leftarrow$
$\quad InputPolicyForOutput(input, policy, output) \land$
$\quad hasPort(output, p) \land hasObligation(policy, ob) \land$
$\quad hasObligationClass(ob, oc) \land hasArgument(ob, arg) \land$
$\quad hasActivationCondition(ob, ac) \land$
$\quad \forall vb.(hasValidityBinding(ob, vb) \rightarrow \exists attr.IOPair(vb, attr, p))$

where the full form of $OutputObligation$ is:

$OutputObligation(oc, arg, ac, p, ob) \equiv$

$\exists oo.A(oo, Obligation) \wedge hasObligationClass(oo, oc)\wedge$

$\exists arg'.hasArgument(oo, arg')\wedge$

$(\forall x.member(arg, x) \rightarrow member(arg', x))\wedge$

$(\forall vb, attr.hasValidityBinding(ob, vb) \wedge IOPair(vb, attr, p)$

$\rightarrow hasValidityBinding(oo, attr))$

where $member(arg, x)$ means $x$ is a member of the list (RDF Collection) $arg$.

