# OpenReview forum: "Perennial Semantic Data Terms of Use for Decentralized Web"
_ACM.org/TheWebConf/2024/Conference — TheWebConf24_

### Official Review · Reviewer_SjLP · 2023-11-24

**Novelty:** 4
**Technical Quality:** 2

**Review:**

The paper proposes the concept of "perennial" Data Terms of Use (DToU) to facilitate users’ control over their data.   Users and applications specify their parts of the DToU policy with local knowledge, covering permissions, requirements, prohibitions, and obligations. Automated reasoning verifies compliance and derives policies for output data. The paper defines
a novel formal description of Data Terms of Use (DToU) and DToU reasoner in a decentralized environment. The proposed solution is based on semantic web technology, facilitating integration with Solid environment.



## strength

- Decentralized web
- Simple solution
- Examples
- Experimental evaluation

## Weakness:

- Scientific problems
- Limitation of the proposal
- State of the art
- language design
- Reproducibility
-Usability studies



## Comments:

Figure 3, which presents general benchmark results, is unreadable.

**Questions:**

Q1: It is unclear what the scientific problem addressed by the paper is. The proposed solution looks like a solution for a specific project rather than a general solution for a scientific problem. What are the general scientific problems?

Q2: From the related research description, it is hard for the reader to link what is written in the detailed approaches and Table 1 (Summarization of features under different categories of related policy languages). What is the relation between Table 1 and the state-of-the-art description?


Q3: The primary purpose of "perennial" Data Terms of Use (DToU) is to facilitate users’ control over their data. As a reader, I expected a user study to show how data and application providers can use the proposal. There is no clear demonstration of how data providers and application providers can utilize this new language and if users find "perennial" Data Terms of Use (DToU) useful.  Therefore, there is a big question about the usability of the proposed concept. Is there any evidence about the usability of DToU ?

Q4: DToU is based on RDF surface, which is not a good choice, as shown in the experimental study. Why RDF surface is chosen ? Is there any alternative ?

Q5: It is written, « It is called “perennial” because a stakeholder only needs to specify the DToU once in the beginning, and it can be reused across activity cycles and stakeholders.»? Again, any evidence about this?


Q6: Is it possible for a user to modify DToU?  What happens in this case for applications that already use DToU?


Q7: The primary motivation of the work is to answer the five requirements in the introduction. However, it is unclear from the language description or the evaluation how these requirements are satisfied.


Q8: What do you mean by « While these approaches vary in terms of expressiveness and usage contexts, they share a common limitation due to the assumption of a finite number of personnel  »?

Q9: What do you mean by «  Just like the plants being implanted once and kept growing in the future »?

**Ethics Review Description:**

NON

**Reviewer Confidence:**

3: The reviewer is confident but not certain that the evaluation is correct

**Scope:**

3: The work is somewhat relevant to the Web and to the track, and is of narrow interest to a sub-community

---

### Official Review · Reviewer_Wd8u · 2023-11-25

**Novelty:** 5
**Technical Quality:** 4

**Review:**

This paper proposes a policy language that can be used to specify and reason with respect to data terms of use in a decentralized web scenario. The proposed DToU policy language is incorporated into the Community Solid Server, which serves as the basis for a performance evaluation.

*Strengths
- The related work section includes a comparative analysis of several policy languages.
- The proposed DToU policy language is integrated into the Community Solid Server.
- The Community Solid Server instantiation of the language is evaluated via a performance benchmark.

*Weaknesses
- The text in the figures is considerably smaller than the article text making them difficult to read on the printed version of the paper.
- The DToU policy language itself and the corresponding reasoning tasks should be formally defined in the body of the paper.
- The evaluation is quite limited as there is no assessment of the expressiveness of the language or the privacy and security guarantees offered by the extended Community Solid Server instantiation.

The work presented is this paper is both relevant and timely, especially considering the various EU directives and regulations in relation to data governance. Despite this potential the current version of the paper is not yet at the standard expected for publication in the proceedings of a top tier conference.

From a presentation perspective the paper is often difficult to digest, partially due to the lack of formal definitions, and partially due to poorly explained and difficult to read figures. The reusability and extendibility of the language could be better supported via formal definitions in addition to the textual descriptions and the examples provided. Additionally, readability could be improved by incorporating the details of the reasoning needed for conformance and obligation checking into the body of the paper.

The proposed benchmark is a nice addition to the paper, however considering the main research artefact is the policy language and the corresponding conformance and obligation checking there is a need to demonstrate that these artefacts are fit for purpose, for instance via an evaluation of the expressiveness and proof of effectiveness of the proposes reasoning. Additionally, there is a need to assess the privacy and security guarantees offered by the extended Community Solid Server instantiation.

**Questions:**

The data provider DToU, application DToU, compliance checking, policy reuse, and data sharing requirements are nicely motivated at the start of the paper and serve as a basis for a comparative evaluation of the literature and for design of the Community Solid Server extensions. However, I’m left wondering about the design of the policy language itself. For instance, what is the basis of the language? How were the concepts derived? Why was a new language created rather than extending an existing language? What is the expressive power of the proposed language?

**Reviewer Confidence:**

4: The reviewer is certain that the evaluation is correct and very familiar with the relevant literature

**Scope:**

4: The work is relevant to the Web and to the track, and is of broad interest to the community

---

### Official Review · Reviewer_gs9K · 2023-11-27

**Novelty:** 5
**Technical Quality:** 5

**Review:**

# Review

## Quality

The paper presents a novel approach to addressing the challenges of data usage management in a decentralized Web context. The authors propose a _perennial_ (as they call it) policy language that separates data policy from application policy, enabling automated compliance checking and policy derivation. The language is built on semantic technologies, which facilitates interoperability and integration with other semantic tools. However, the authors fail to properly justify why they are "reinventing the wheel" instead of adopting languages like ODRL for modelling their policies. The paper also provides an evaluation and additional axioms as an appendix.

## Clarity

The paper is well-structured and easy to understand, with mostly clear explanations of the proposed concepts and methodology. The authors provide concise definitions of key terms and use examples to illustrate their points.

## Originality

The paper introduces an approach to data usage management in a decentralized Web context. The authors' use of semantic technologies to express and reason about policies is relevant, but as mentioned earlier, the proposed policy language (imo) reinvents the wheel instead of building upon (or extending) existing languages/standards like ODRL.

## Significance

The paper addresses a critical challenge in the development of decentralized Web applications: how to ensure that data is used in a responsible and compliant manner.

## Pros & Cons

 | Pros                                                                                                      | Cons                                                                    |
 | --------------------------------------------------------------------------------------------------------- | ----------------------------------------------------------------------- |
 | Interesting approach to addressing the challenges of data usage management in a decentralized Web context | Proposed language may be too complex for some users.                    |
 | Additional axioms provided as supplemental material in the Appendix                                       | Dismissal of existing languages like ODRL wasn't sufficiently motivated |
 | Addresses a critical challenge in the development of decentralized Web applications                       |                                                                         |


# Detailed comments

In the following

- ⩝ x ... is used to indicate that x is missing
- ⇋ x ... substitute the highlighted text with x


##  1 Introduction

---

- >`[p.1, 98-99]`: Data Terms of Use (DToU), a formal language model **that targets at**
  >`[p.1, 99-100]`: **addressing** the following challenges in a decentralized Web context:
  - ⇋ that addresses
- >`[p.1, 100-101]`:  2) imparting application’s (developer’s) DToU on how they handle data;
  - what does that mean? please elaborate
- >`[p.1, 109-110]`: called **“perennial”** because a stakeholder only needs to specify the
  - a more commonly used term for e.g. licenses that have no enddate is _perpetual_
- >`[p.1, 111-112]`: **across activity cycles and across stakeholders**, just like the plants being **implanted**
  - what's an _activity cycle_?
  - ⇋ planted

##  2 Related Research

---

- >`[p.2, 173-174]`: supposes the presence of a **‘supervisor’** with abundant knowledge
  - == Trusted 3rd party?
- >`[p.2, 176-177]`: are **personnel changes**. While this assumption is reasonable for
  - what?
- >`[p.2, 191-192]`: itation due to the assumption of **a finite number of personnel**. In
  - what? what does that mean?
- >`[p.2, 199-200]`: rules for policy evolution. Additionally, it uses custom **meta-code**
  - what's _meta-code_?
- >`[p.2, 201-202]`: **most expressive part, the meta-code,** is arbitrary and difficult to stat-
  - of what? the policy expression?
- >`[p.2, 202-203]`: **ically verify.** CamFlow [25], on the other hand, borrows concepts
  - why? according to whom? proof?
- >`[p.2, 227-228]`: ODRL primarily focuses on expressing what is (not) permitted for the data, and **lacks a corresponding mechanism for expressing application information**.
  - such as? if such information can be expressed with any other kind of ontology/vocabulary it's not necessary for ODRL to be able to express this itself, right?


- >`[p.3, 291-292]`: Table 1: Summarization of features under different categories of related policy languages.
  - What's the + in [XACML/ODRL][C1] = AO+ ?
  - generally, it would improve the readability of Table1 tremendously, if you wouldn't use the same symbol for different concepts across columns, e.g. T means true in C3 and C4 but Transformation in C5.

###  3.1 Language design

---

- >`[p.3, 331-332]`: expressed in the attribute layer. In the **Turtle** syntax of the policy
  - that should be the same in any RDF serialisation, right?
- >`[p.3, 334-335]`: More precisely, each **attributes** is a simple tuple: (name, class,
  - ⇋ attribute
  - your tuples remind me of RDF reification.. are there similarities?
- >`[p.3, 345-346]`: of “alice@b.c.”
  - ⇋ "alice@b.c".

- >`[p.4, 372-373]`: the requirements or available resources. Two typical **types of tags**
  >`[p.4, 374-375]`: are **security** and integrity: security specifies which security level(s)
  - according to Fig1, Security isn't a Tag, but a Requirement.
- >`[p.4, 380-381]`: class of an attribute using attribute_ref; **other information in the**
  >`[p.4, 382-383]`: **attribute are not used by a tag**. Our policy language also allows
  - what?
- >`[p.4, 389-390]`: `:value :nil.`
  >`[p.4, 392-393]`: `:value :nil.`
  >`[p.4, 396-397]`: `:value :nil.`
  - what's the purpose of explicitly "nulling" the value? circumventing OWA?

- >`[policies]`: all policy examples
  - have you actually tried to use ODRL for modelling your policies? on a first glance I would say it supports expressing almost all (if not all) concepts you are modelling while at the same time being more comprehensible and easier to read/understand.
  - alternatively, maybe as an ODRL profile?


##  5.2 Results and discussions

---

- >`[p.8, 821-822]`: Figure 3,4,5:
  - please add units to the axis labels

##  A.2 Conformance check

---

- >`[p.10, 1122-1123]`: ∃𝑥.𝐴(𝑥,𝑈𝑛𝑠𝑎𝑡𝑖𝑠𝑓𝑖𝑒𝑑𝑅𝑒𝑞𝑢𝑖𝑟𝑒𝑚𝑒𝑛𝑡) ∧ ℎ𝑎𝑠𝑇𝑦𝑝𝑒(𝑥,𝑡)∧
  - so what's hasType? I thought `𝐴(𝑥,𝑈𝑛𝑠𝑎𝑡𝑖𝑠𝑓𝑖𝑒𝑑𝑅𝑒𝑞𝑢𝑖𝑟𝑒𝑚𝑒𝑛𝑡)` states that type of x is `𝑈𝑛𝑠𝑎𝑡𝑖𝑠𝑓𝑖𝑒𝑑𝑅𝑒𝑞𝑢𝑖𝑟𝑒𝑚𝑒𝑛𝑡`?

**Questions:**

see above

**Reviewer Confidence:**

4: The reviewer is certain that the evaluation is correct and very familiar with the relevant literature

**Scope:**

4: The work is relevant to the Web and to the track, and is of broad interest to the community

---

### Official Review · Reviewer_mmsn · 2023-11-29

**Novelty:** 7
**Technical Quality:** 6

**Review:**

The paper proposes a formal model for reasoning about Data Terms of Use (DToU), using RDF Surfaces (essentially, a first-order logic extension of RDF), to support data usage policy negotiation and enforcement in a decentralised context. In particular, the idea is to allow individuals to specify DToU at a general rather than application-specific level, and have compliant applications negotiate data usage with, e.g., their Solid pod, with minimal user effort or technical know-how.

Pros
==
- The problem addressed by the paper is an increasingly important one
- The proposal is timely and well thought-through.
- I believe it is extremely relevant to this conference.
- The evaluation made sense and was largely convincing.

Cons
==
- I found Table 1 a little confusing - I had to go looking to find the challenges C1 to C5; they aren't referred to like this when they're introduced in the text (they're just numbered 1-5).
- The values in the table are also never explained: "For column C2, C means capacity, I means multi-input, M means use mode", etc., is only meaningful to someone who knows what capacity, multi-input, use mode mean in the context of "imparting application's DToU on how they handle data" (C2). I'd appreciate this being clarified.
- Section 3.1 (Language design) might benefit from a bit more of the motivation for particular design choices. What's behind the attribute/semantics distinction? What's the general concept that "tag" is aiming to capture?, etc.
- It's unclear how the test data and app policies were created, or how much (if at all) their contents reflect plausible actual policies, and it's quite possible that overall performance might be different (positively or negatively) on realistic test policies.

Minor aside: is it important to draw a distinction between different concrete RDF syntaxes? Notation3, JSON-LD, Turtle, etc., are all designed to be intertranslatable save in a very few (rare) JSON-LD corner cases. RDF-expressibility seems like the relevant property of a DToU language here, unless I'm missing something?

**Questions:**

- What do the values in table 1 mean (e.g., capacity, multi-input, use mode, etc.) in the context of the relevant challenges?
- How were the test data and app policies created, and how much do/might they reflect real-world policies?
- Is it important to draw a distinction between different concrete RDF syntaxes?
- Any thoughts/responses about the other points I raised in the review?

**Reviewer Confidence:**

3: The reviewer is confident but not certain that the evaluation is correct

**Scope:**

4: The work is relevant to the Web and to the track, and is of broad interest to the community

---

### Official Review · Reviewer_peB5 · 2023-12-01

**Novelty:** 5
**Technical Quality:** 6

**Review:**

The authors propose a formal semantic policy language for describing the Data Terms of Use (DToU) in the context of decentralized Web. Turtle, Notation 3, and RDF Surfaces are made use of here. This language has been integrated with the Solid framework. Evaluation is performed by varying the values of key variables.

Strengths
1) Relevant to the conference and the track.
2) This is an important area of work, and the policy language could be useful to the community.
3) The paper is well-written.

Weaknesses
1) The research component is not clear.
2) Real-world use cases are not used for evaluation.
3) Comparison with other policy languages, such as ODRL and AIR, is missing.

Other comments/typos
1) Figure 1; instead of Security being a subclass of Requirement, it would be better to call the Security class as SecurityRequirement.
2) Line 599, space is missing after distinct.
3) What does the value in the X-axis mean in Figures 3, 4, and 5?

**Questions:**

1) The policy language consists of a vocabulary defined using N3 and a reasoner. The research problem that the authors are trying to solve is not clear. What is it?
2) Figure 1; what motivated this hierarchy/relationships between the concepts? Are there any use cases that you looked at? If you did, why aren't they used in the evaluation?
3) Table 1; where are the challenges and the terms used in the table (A, O, T, etc.) described?
4) Line 675; the terms benchmark and policy language have been used interchangeably. Why is this a bechmark?

**Reviewer Confidence:**

3: The reviewer is confident but not certain that the evaluation is correct

**Scope:**

4: The work is relevant to the Web and to the track, and is of broad interest to the community

---

### Decision · Program_Chairs · 2024-01-22

**Decision:**

Accept

**Comment:**

This paper proposes a formal model for addressing the challenges of reasoning with Data Terms of Use in the decentralised web. This approach designs a policy language that is represented using RDF surfaces to support the representation and enforcement of the policies.The policy language proposed eparates data policy from application policy, enabling automated compliance checking and policy derivation.This language is already included in the Community Solid Server, therefore giving good prospects for future impact.

 The proposed approach is novel and the paper is generally well written and is significant for users interested in policy languages.


 Pros

 Relevant and topical problem
 Solid evaluation
 Additional axiomatisation provided in the supplemental material
 The proposed policy language is included in the Community Solid Server

 Cons

 Table 1 is confusing and needs clarification

 Clarify better the motivation of the proposed approach wrt adaptation and extensions of current standards such as ODRL

 Lack of formal definitions makes the paper somewhat difficult to read